# Prevalence of congenital missing permanent teeth and its association with side and gender in a Saudi subpopulation

Abdulrahman K. Alshammari[1]*, Muteb A. Algharbi[1], Freah L. Alshammary[1], Nabeel S. Almotairy[2], Hatem D. Alshammari[1], Ibrahim R. Altheban[3], Khalid A. Aljameel[3], Fahad M. Aldakheel[3], Ahmed A. Madfa[4]

**1** Department of Preventive Dentistry, College of Dentistry, University of Ha'il, Ha'il, Kingdom of Saudi Arabia, **2** Department of Orthodontics and Pediatric Dentistry, College of Dentistry, Qassim University, Buraidah, Saudi Arabia, **3** Dental Intern, College of Dentistry, University of Ha'il, Ha'il, Kingdom of Saudi Arabia, **4** Department of Restorative Dental Science, College of Dentistry, University of Ha'il, Ha'il, Kingdom of Saudi Arabia

* abra.alshammari@uoh.edu.sa

## Abstract

### Background

This study's objectives are to ascertain the frequency of congenital tooth missing and evaluate the relationship between gender and side characteristics.

### Method

This retrospective study involved participants who attended private dental clinics as well as a dental college in Hail, Saudi Arabia. Out of the 1,150 patients examined, 494 (220 males and 274 females) fulfilled the inclusion criteria. Different types of tooth agenesis were evaluated and classified into the following categories: mild (one or two teeth missing), moderate (three to five teeth missing), and severe (six or more teeth missing). The analysis did not include third molars. The existence of retained primary teeth was noted, along with any dental abnormalities accompanying tooth agenesis. Statistical analysis was conducted utilizing the chi-square test and Fisher's exact test to explore possible relationships between variables. A significance level of 5% (alpha = 0.05) was utilized.

### Results

Out of the 1150 panoramic radiographs that were examined, 494 individuals (220 (44.5%) males and 274 (55.5%) females) met the criteria for inclusion. The prevalence of congenitally missing permanent teeth was 65 (13.2%) in the study sample. The prevalence of congenitally missing teeth were more incidence in maxilla 36 (56.3%) than mandible 28 (43.8%). There is statistically significant association between the occurrence of congenitally missing teeth and gender or arch ($p < 0.05$).

Data availability statement: All relevant data are within the paper and its Supporting Information files.

Funding: This research has been funded by Scientific Research Deanship at University of Ha'il - Saudi Arabia through project number <RG-23 113>. The funder had no role in study design, data collection and analysis, decision to publish, or preparation of the manuscript.

Competing interests: The authors have declared that no competing interests exist.

The prevalence of dental anomalies was similar in both sides. There are not a statistically significant association ($p > 0.05$) between the occurrence of congenitally missing teeth and side. The second premolar was the most commonly missing (7.3%). The retained deciduous teeth was shown to be the most common dental anomaly, with a prevalence of 15 (23.4%).

## Conclusions

The prevalence of congenitally missing teeth fell within the range reported in previous studies. Second premolars were the most frequently congenitally missing teeth, with maxillary teeth more commonly affected than mandibular ones.

## Background

The development of the human dentition is a complicated process that involves both biological and genetic components [1]. Any flaw in this process can cause dental anomalies, which are differences in the shape, location, size, and quantity of teeth, and can cause dental problems or impacts the dental development [2,3]. A common dental anomaly is tooth agenesis (hypodontia), which is a congenital anomaly characterized by the absence of one or more teeth in the jaws [4]. The presence of tooth agenesis can cause functional and aesthetic defects, including malocclusion of the teeth, periodontal impairment, and reduced alveolar bone level, which emphasizes the importance of early intervention and treatment [5]. Additionally, addressing these anomaly defects is crucial because they might impede the treatment and result in problems with tooth malalignment, dental arch, and teeth eruption.

Agenesis is believed to have a multifactorial origin. It can be caused by genetic or environmental factors, or both may be responsible for its occurrence, where genetic differences have been shown to be the main factors leading to this condition [6]. Congenitally missing teeth have been associated with a variety of local and systemic environmental factors, including squeezed germ formation in the affected area, dental lamina ruptures, lack of space in the area, abnormalities in the function of the dental epithelium, issues with the formation of mesenchymal tissue, and ruptures during the embryonic fusion of the upper jaw and medial nasal process [7]. Congenital tooth missing has also been associated with hormonal changes, infections, radiation, trauma [8]. Dental agenesis is sometimes a symptom of more common systemic diseases, including Down syndrome, ectodermal dysplasia, and cleft lip and palate [8,9].

Other dental defects such as tooth impaction, ectopic eruption, and microdontia frequently coexist with tooth agenesis [10]. Ben-Bassat and Brin report that patients born with tooth agenesis exhibit specific craniofacial morphology and growth patterns including prognathic mandible, a smaller angle of the mandibular plane, shorter maxilla, and a retroclination of the mandibular and maxillary incisors [11]. These characteristics were found to be influenced by the severity of congenitally missing teeth and are most likely the result of the underdevelopment of the apical base as a result of the absence of tooth buds. Furthermore, dental agenesis has been found to cause problems with a person's ability to speak, aesthetics, and use muscle's function [12].

Numerous studies have examined the prevalence of tooth agenesis in various global populations, with results ranging from 2.7% to 14.7%. [13–15]. However, the prevalence of tooth agenesis in Saudi Arabia has been reported to range from 4% to 31.68%, where agenesis has been shown to be one of the most common dental anomalies in Saudi Arabia after impaction [16–22]. The prevalence of agenesis in a study conducted in central Saudi Arabia was 4% and the tooth most frequently affected by agenesis was the mandibular second premolar, the maxillary lateral, and the second maxillary premolar [16]. However, another report from central Saudi Arabia found that the prevalence of agenesis was (12.5%) and was most frequently associated with infraocclusion [23]. Sajjad et al. [24] conducted in northern Saudi Arabia reported that the prevalence of agenesis was 6.1% with most participants had one or two congenitally missing teeth with women have higher prevalence than men. Furthermore, they found no significant differences between the sides with the most commonly affected teeth with agenesis were the mandibular second premolar (40.1%), followed by the maxillary lateral incisors (20.4%) and then the second maxillary premolars (12.6%). On the other hand, a higher prevalence of agenesis was reported in south of Saudi Arabia (21.2%) with a significant difference between genders (p < 0.05) [25]. Variations in the diagnostic criteria, sample techniques, and ethnic makeup of the research may be the cause of these inconsistent findings.

It is concerning that the Hail district, which is in the northern area, has not been the subject of a thorough study to determine the prevalence of congenital tooth loss in its people. Therefore, it is crucial to evaluate the frequency of congenital tooth loss and determine the gender and association side of this condition. Since doing so would make it possible for dental practitioners to carry out preventive and restorative management, which could ultimately significantly enhance the quality of life associated with oral health (OHRQoL). Therefore, the objective of this study was to determine the prevalence of congenitally missing teeth and to assess the association between the side and the gender factors. By applying this knowledge, experts and public health practitioners will be capable of establishing protocols and devising preventive strategies for the interceptive management of congenitally missing teeth in the Hail province.

## Methods

### Study design

The retrospective study sample consisted of individuals who attended a private dental clinic in Hail, Saudi Arabia, and the Clinics in College of Dentistry at Hail University between July 2019 and June 2024. The primary check records were digital panoramic radiographs and intraoral photographs; cephalometric radiographs and study casts were available to the researchers upon request. Three calibrated examiners (IRA, KAA, and FMA) studied 1150 dental patients aged 9−60 years. Ethical approval for this study was obtained from the Hail University Research Committee (approval number H-2023–349). This study was conducted in accordance with the protocol set forth in the Declaration of the World Medical Association.

### Sample size calculation

The sample size was calculated using Cochran's formula: $N = (Z_\alpha 2 \times P (1-P))/D2$. Here, $Z_\alpha$ represents the critical value of the normal distribution at $\alpha/2$ (1.96), D represents the desired degree of precision, and P represents the prevalence of dental anomalies (80%) determined by a published study in the eastern region of Saudi Arabia [20]. The recommended sample size for the investigation was 228. The final sample consisted of 494 participants.

### Sample selection

Patients needed to have sufficient orthodontic records and be at least 9 years old to be recruited in the study. The lower age limit of the sample was reasonable for assessing agenesis in all permanent teeth (except the third molars). Individuals who had many permanent teeth extracted, had a history of orthodontic therapy, had insufficient orthodontic records, had craniofacial syndrome or cleft lip and palate, or were receiving prosthetic treatment were not included. 494 (220 males and 274 females) of the 1150 patients satisfied the requirements for inclusion.

## Outcome assessment and statistical analysis

Various types of tooth agenesis were evaluated. Agenesis was classified as mild, involving one or two missing teeth, moderate, involving three to five missing teeth, and severe, involving six or more missing teeth. Third molars were not included in the analysis. Retained deciduous teeth were reported, and some abnormalities coexisted with tooth agenesis also were recorded in this study. In ideal lighting conditions, three experienced assessors used standard screen brightness and resolution to analyze orthodontic data.

The examiner received calibration training before performing the assessment. Calibration processes were overseen by AKA, MAA, FLA, NSA, and AAM, each possessing more than ten years of experience. The examiner—IRA, KAA, and FMA—took part in the calibration procedure, adhering to set guidelines for standardization and variance before the initiation of experimental measurements. A sample of 20 cases was randomly chosen for assessment. The level of inter-observer agreement was assessed using the kappa coefficient. Observers collaboratively examined and deliberated on inconsistencies to achieve agreement. Two weeks following the initial assessment, the same evaluator performed a second, blinded review, having no access to the first findings. Intra-observer reliability was evaluated using 20 samples. The level of inter-observer and intra-observer agreement was determined to be 91 and 94%, respectively.

IBM Co.'s Statistical Package for the Social Sciences, version 22 (SPSS), was used to analyze the study's data. Numerous statistical operations, such as frequency distribution and cross-tabulation, were carried out using this software. An analysis of the frequency of dental abnormalities was part of the study. Gender, type of dental arches, and locations were among the several variables reported in this study. The chi-square and Fisher's exact tests were used to assess any possible correlations between these variables. At 5% (Alpha = 0.05), the significance threshold was established.

## Results

Out of the 1150 panoramic radiographs that were examined, 494 individuals (220 (44.5%) males and 274 (55.5%) females) met the criteria for inclusion. The prevalence of congenitally missing permanent teeth was 65 (13.2%) in the study sample. Fig 1 displays the frequency of congenitally missing teeth. The association between the congenitally missing teeth and various variables are reported in Table 1. Overall, missing teeth were found more commonly in females 41 (64.1%) than in males 23 (35.9%). There is statistically significant association between the occurrence of congenitally missing teeth and gender ($p < 0.05$). Maxilla had a higher incidence of congenitally missing teeth 36 (56.3%) than mandible 28 (43.8%). A statistically significant ($p < 0.05$) correlation between the prevalence of congenitally missing teeth and dental arches. On both sides, the prevalence of congenitally missing was comparable. The frequency of dental congenitally missing teeth and side effects do not statistically significant ($p > 0.05$).

The frequency of various congenitally missing teeth is shown in Fig 2 and Table 2 according to gender, side, and dental arch. The most common congenitally missing teeth were the second premolar (35), which was followed by the first premolar (12), canine (8), and lateral incisors (7). Gender and the type of congenitally missing teeth are statistically significant ($p < 0.05$). Furthermore, there is a statistically significant ($p < 0.05$) between dental arches and the type of congenitally missing teeth. The frequency of congenitally missing teeth and side effects do not statistically significant ($p > 0.05$).

The frequency of various dental abnormalities linked to congenitally missing teeth is shown in Table 2 by dental arch, side, and gender. With a prevalence of 15 (23.4%), retained deciduous teeth were found to be the most prevalent dental anomaly in this study. Females were shown to experience it more frequently than males. The maxillary showed the highest frequency of retained deciduous teeth. Impaction was shown to be the second most prevalent dental anomaly in this study, occurring in 11 (17.2%) of all cases. females were more likely than male to have it. Additionally, the left side of the maxilla showed the most impaction. In this examination, rotation was shown to be the third most common dental anomaly, accounting for two (3.1%) of the examined cases. Compared to boys, girls were more likely to experience this phenomenon. On the mandibular side, the most frequent rotation was also seen. Apical resorption linked to congenitally missing teeth was more frequently observed apically in 21.9% of cases. females were more likely than males to have it.

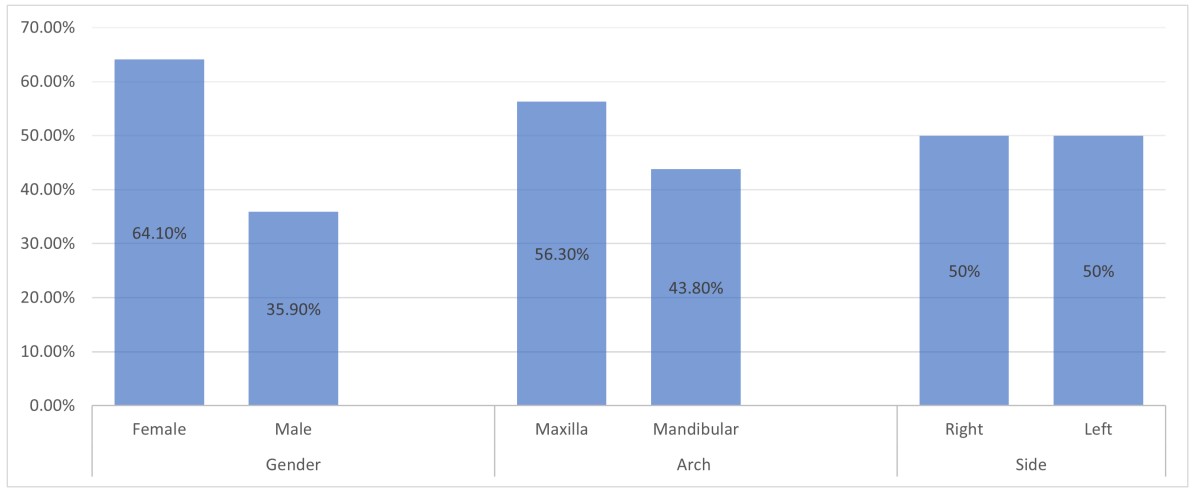

**Fig 1. Frequency of congenitally missing teeth by gender, dental arch, and side.**

**Table 1. Distribution congenitally missing teeth according to gender, side and dental arch.**

| Variables | | Congenitally missing teeth | p-value |
|---|---|---|---|
| **Gender** | Male | 23 (35.9%) | <.05 |
| | Female | 41 (64.1%) | |
| **Dental arch** | Maxilla | 36 (56.3%) | <.05 |
| | Mandible | 28 (43.8%) | |
| **Side** | Right | 32 (50%) | >0.05 |
| | Left | 32 (50% | |

As indicated in Table 2, the most commonly seen deciduous tooth that retained in place was also located on the right side of the maxilla. Overall, mild agenesis predominated, particularly in females and the maxillary arch; moderate agenesis was less common, and severe agenesis was not found. However, The severity of congenitally missing teeth and gender, arch or side do not statistically significant (p>0.05).

## Discussion

The most common dental anomaly, congenitally missing teeth, can cause both functional and esthetic problems, necessitating a costly and intricate multidisciplinary treatment strategy [26]. Early orthodontic therapy was greatly aided by the diagnosis of agenesis. Alternative treatment options might be planned and carried out using multidisciplinary team tactics if the indication of a congenitally missing tooth was identified prior to the occlusion stabilizing [27,28]. Dental professionals should be aware of the detrimental consequences on the occlusion of congenitally missing teeth, as they are a common developmental abnormality [29]. In order to determine the quantity and location of permanent missing teeth, this study examined agenesis in orthodontic patients' pre-treatment files. We found that 13.2% of the Saudi subgroup examined had agenesis. Aside from third molars, the occurrence of agenesis was similar to the results from studies by Altan et al. [30] and Musaed et al. [31], which showed that the prevalence of permanently missing teeth was 14.1% and 16.3%, respectively. Sisman et al. [27] investigated the permanently missing teeth in a group of Turkish orthodontic patients and discovered that agenesis occurred less frequently in their sample (7.54%) compared to ours. However, the results reported in this study were higher than those observed in other populations, including 6.3% in the Brazilian population [32], 8.5% in

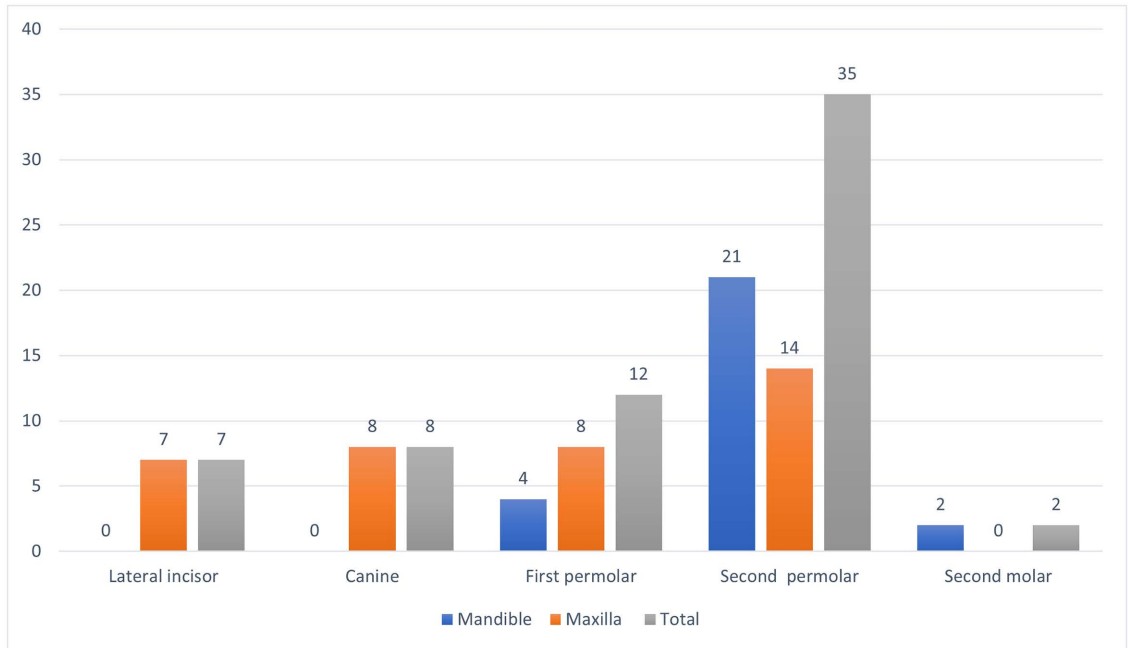

**Fig 2. Frequency of congenitally missing teeth by tooth type.**

**Table 2. Distribution type congenitally missing teeth and dental anomalies associated with missing teeth according to gender, side, and dental arch.**

| Variables | | Gender | | Arch | | Side | |
|---|---|---|---|---|---|---|---|
| | | Male | Female | Maxilla | Mandible | Right | Left |
| Congenitally missing teeth | Lateral | 5(7.8%) | 2(3.1%) | 7(10.9%) | 0 | 3(4.7%) | 4(6.3%) |
| | Canine | 3(4.7%) | 5(7.8%) | 8(12.5%) | 0 | 3(4.7%) | 5(7.8%) |
| | 1st premolars | 3(4.7%) | 9(14.1%) | 8(12.5%) | 4(6.3%) | 6(9.4%) | 6(9.4%) |
| | 2nd premolars | 12(18.8%) | 23(35.9%) | 13(20.3%) | 22(34.4%) | 19(29.7%) | 16(25%) |
| | 2nd molar | 0 | 2(3.1%) | 0 | 2(3.1%) | 1(1.6%) | 1(1.6%) |
| | *P-value* | **.076** | | **<0.00** | | **.447** | |
| Dental anomalies associated with missing teeth | Impaction | 2(3.1%) | 9(14.1%) | 6(9.4%) | 5(7.8%) | 4(6.3%) | 7(10.9%) |
| | Retained deciduous teeth | 9(14.1%) | 6(9.4%) | 8(12.5%) | 7(10.9%) | 8(12.5%) | 7(10.9%) |
| | Rotation | 0 | 2(3.1%) | 0 | 2(3.1%) | 1(1.6%) | 1(1.6%) |
| | *P-value* | **.128** | | **.339** | | **.880** | |
| Degree of root resorption | No | 2(3.1%) | 7(10.9%) | 5(7.8%) | 4(6.3%) | 5(7.8%) | 4(6.3%) |
| | Apical | 6(9.4%) | 8(12.5%) | 7(10.9%) | 7(10.9%) | 7(10.9%) | 7(10.9%) |
| | Middle | 3(4.7%) | 3(4.7%) | 4(6.3%) | 2(3.1%) | 3(4.7%) | 3(4.7%) |
| | *P-value* | .125 | | .988 | | .907 | |
| Severity tooth agenesis | Mild | 15 (23.4%) | 31 (48.4%) | 28 (43.8%) | 18 (28.1%) | 21 (32.8%) | 25 (39.1%) |
| | Moderate | 8 (12.5%) | 10 (15.6%) | 8 (12.5%) | 10 (15.6%) | 11 (17.2%) | 7 (10.9%) |
| | *P-value* | **.273** | | **181** | | **.202** | |

the Japanese population [33], 10.1% in the Norwegian population [34], and 11.3% in the Irish population [35]. On the other hand, our study reported a lower prevalence of agenesis compared to the findings of Scheiwiller et al. [36], Sheikhi et al. [37], and Arif et al. [38], who reported rates of 50.8%, 45.7%, and 33.3%, respectively.

Out of the 494 patients, 44.5% were male, while 55.5% were female. In the present study, the occurrence of congenitally missing teeth was 35.9% in males and 64.1% in females. The present study found that agenesis occurred more frequently in women than men. Earlier research has consistently documented this [27]. This happened simultaneously with the studies published by Sisman [27] and Kathariya et al. [39]. The occurrence of agenesis was observed to be more common in boys compared to girls, based on varying information from Arif et al. [38] and Schonberger et al. [40]. Environmental factors may change biological differences, including a smaller jaw, which could explain the higher incidence seen in females. According to some theories, this does not necessarily indicate that agenesis is more common in women, but rather that women are more likely to seek therapy because they care more about appearance. The gender bias of persons attending the clinic likewise reflected this. Although this may seem plausible, several earlier studies have shown that gender has no association with a desire to seek orthodontic treatment [41,42,43].

The research indicated that the maxillary arch exhibited a higher percentage of congenitally missing teeth (56.3%) compared to the mandibular arch (43.8%). According to Schonberger et al. [40] and Chib et al. [44], the occurrence of congenitally missing teeth was higher, showing 56.8% overall, 55% in the maxilla, and 43.2% in the mandible. In the maxillary arch, Kapdan et al. [45] and Vahid et al. [46] observed a higher prevalence of congenitally missing teeth. Conversely, Chung et al. [47] and Hassan et al. [48] found that more congenitally missing teeth were identified in the mandible compared to the maxilla.

The pattern of congenitally missing teeth in the current investigation with bilateral agenesis did not differ statistically. The results concurred with those of Uzuner et al. [49], who found no statistically significant difference between the two groups. When the number of missing teeth on the left and right sides were taken into account, Rakshan et al. [50] found no significant difference. Second premolar agenesis was the most common symmetric agenesis seen, followed by lateral incisor or first premolar agenesis. The results were consistent with earlier research [33,51].

Retained deciduous teeth were reported, and some abnormalities coexisted with tooth agenesis. The outcomes matched those of Musaed Ziad et al. [31] Its existence may result in developmental problems that raise functional and aesthetic challenges. In the study, congenitally missing teeth were also linked to the prevalence of impaction, which was followed by rotating teeth. This was consistent with the findings of Bakhurji et al. [21]. When evaluating patients, this information would help doctors recognize this prevalent problem and better prepare them to handle patients. Improving people's general health and averting any consequences requires increasing awareness, encouraging early detection, and offering easily accessible dental care.

## Clinical implications

The results of this research provide an important understanding of the epidemiology of congenitally missing permanent teeth in a Saudi subpopulation. Grasping the prevalence and trends of tooth agenesis can assist clinicians in diagnosis, treatment planning, and strategies for early intervention. The lack of notable side-based variations implies that agenesis might be determined more by other genetic or environmental influences than by lateral symmetry.

From a public health viewpoint, recognizing trends specific to populations supports resource allocation and the creation of preventive and therapeutic dental initiatives designed for community requirements. Timely identification, especially among high-risk populations like women or those with a family history, could enable more efficient interceptive orthodontics and prosthetic treatment. Furthermore, the research highlights the significance of standardizing the clinical evaluation and documentation of agenesis, which could enhance communication between professionals and lead to improved long-term outcomes for patients. Additional studies investigating genetic, environmental, and systemic factors affecting tooth agenesis in local populations are needed to enhance these findings.

The study's limitations include the possibility of selection bias because of the small sample size, the use of self-reported data and orthopantomograms, the failure to account for genetic and confounding variables, and the cross-sectional design, which could affect the study's validity and generalizability. One of the study's limitations was that it only used data from one location. A multi-center approach should be the goal of future research in order to better represent the Saudi populace. The respondents' incapacity to remember prior oral events (extractions or trauma) is another drawback of the study's retrospective design, particularly if those experiences happened earlier in life or were treated at a different dental clinic. The retrospective design inherently restricts the ability to manage data quality and uniformity. Dependence on current clinical and radiographic records might have led to information bias, especially if instances of congenitally missing teeth were underreported or poorly documented between clinics. Lacking or unfinished patient information might also impact the precision of prevalence calculations. Selection bias poses another issue since the study sample was obtained from people visiting dental clinics and a dental college, potentially making it unrepresentative of the general population. Individuals seeking dental treatment are more prone to exhibit clinical issues, possibly inflating the perceived occurrence of tooth agenesis in comparison to the larger, untreated demographic. Lastly, the sample age was range from 9 to 60 years, and this one of the limitations of the study since there were large variability in tooth development and eruption times between population. However, we did not classify the age group since agenesis will not be affected by younger or adult's patient.

## Conclusions

The prevalence of congenitally missing teeth were within the range reported previously. The second premolars were the most commonly missing teeth. Maxillary teeth were more prone to be missing. The incidence of agenesis was unaffected by the occurrence's sides. Dental agenesis was more prevalent in females. However, agenesis appeared to be more significant in males in terms of treatment required because males had a higher number of congenitally missing teeth per patient. Dental practitioners will be better able to understand tooth agenesis thanks to the information this study provides. With this knowledge, they would be able to develop treatment plans that satisfy both functional and aesthetic needs, improving treatment quality in the process.

## Supporting information

**S1 Data.  Include the protocols of the study and all the data used for analysis.**
(XLSX)

## Author contributions

**Conceptualization:** Abdulrahman K. Alshammari, Muteb A. Algharbi, Freah L. Alshammary, Nabeel S. Almotairy, Hatem D. Alshammari, Ibrahim R. Altheban, Ahmed A. Madfa.

**Data curation:** Abdulrahman K. Alshammari, Muteb A. Algharbi, Freah L. Alshammary, Nabeel S. Almotairy, Hatem D. Alshammari, Ibrahim R. Altheban, Khalid A. Aljameel, Fahad M. Aldakheel, Ahmed A. Madfa.

**Formal analysis:** Abdulrahman K. Alshammari, Muteb A. Algharbi, Freah L. Alshammary, Nabeel S. Almotairy, Hatem D. Alshammari, Ahmed A. Madfa.

**Funding acquisition:** Abdulrahman K. Alshammari, Muteb A. Algharbi, Freah L. Alshammary, Nabeel S. Almotairy, Hatem D. Alshammari, Ahmed A. Madfa.

**Investigation:** Abdulrahman K. Alshammari, Muteb A. Algharbi, Freah L. Alshammary, Nabeel S. Almotairy, Hatem D. Alshammari, Ibrahim R. Altheban, Khalid A. Aljameel, Fahad M. Aldakheel, Ahmed A. Madfa.

**Methodology:** Abdulrahman K. Alshammari, Muteb A. Algharbi, Freah L. Alshammary, Nabeel S. Almotairy, Hatem D. Alshammari, Khalid A. Aljameel, Fahad M. Aldakheel, Ahmed A. Madfa.

**Project administration:** Abdulrahman K. Alshammari, Muteb A. Algharbi, Freah L. Alshammary, Nabeel S. Almotairy, Hatem D. Alshammari, Ahmed A. Madfa.

**Resources:** Abdulrahman K. Alshammari, Muteb A. Algharbi, Freah L. Alshammary, Nabeel S. Almotairy, Hatem D. Alshammari, Ahmed A. Madfa.

**Software:** Abdulrahman K. Alshammari, Muteb A. Algharbi, Freah L. Alshammary, Nabeel S. Almotairy, Hatem D. Alshammari, Ahmed A. Madfa.

**Supervision:** Abdulrahman K. Alshammari, Muteb A. Algharbi, Freah L. Alshammary, Nabeel S. Almotairy, Hatem D. Alshammari, Ahmed A. Madfa.

**Validation:** Abdulrahman K. Alshammari, Muteb A. Algharbi, Freah L. Alshammary, Nabeel S. Almotairy, Hatem D. Alshammari, Ahmed A. Madfa.

**Visualization:** Abdulrahman K. Alshammari, Muteb A. Algharbi, Freah L. Alshammary, Nabeel S. Almotairy, Hatem D. Alshammari, Khalid A. Aljameel, Fahad M. Aldakheel, Ahmed A. Madfa.

**Writing – original draft:** Abdulrahman K. Alshammari, Muteb A. Algharbi, Freah L. Alshammary, Nabeel S. Almotairy, Hatem D. Alshammari, Ahmed A. Madfa.

**Writing – review & editing:** Abdulrahman K. Alshammari, Muteb A. Algharbi, Freah L. Alshammary, Nabeel S. Almotairy, Hatem D. Alshammari, Ahmed A. Madfa.

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
