## [Decision Letter · Decision Letter 0]

16 May 2025

PONE-D-25-02469Prevalence of congenital missing permanent teeth and explore any correlation with side and gender in a sample from Saudi subpopulationPLOS ONE

Dear Dr. Alshammari,

Thank you for submitting your manuscript to PLOS ONE. After careful consideration, we feel that it has merit but does not fully meet PLOS ONE’s publication criteria as it currently stands. Therefore, we invite you to submit a revised version of the manuscript that addresses the points raised during the review process.

Please review and address concerns from the reviewers especially in the methodology and the high prevalent of third molar agenesis explanation. ==============================

We look forward to receiving your revised manuscript.

Kind regards,

Sompop Bencharit, DDS, MS, PhD, FACP

Academic Editor

PLOS ONE

Journal Requirements:

[This research has been funded by Scientific Research Deanship at University of Ha’il - Saudi Arabia through project number <<RG-23 113>>]. 

4. Please include a copy of Tables 1, 2, and 3 which you refer to in your text on pages 7, 8, 9.

Additional Editor Comment:

The reviewers have diverse opinion of this work. Please carefully review and address their concerns.

Reviewers' comments:

Reviewer's Responses to Questions

**Comments to the Author**

1. Is the manuscript technically sound, and do the data support the conclusions?

Reviewer #1: No

Reviewer #2: Yes

2. Has the statistical analysis been performed appropriately and rigorously? 

Reviewer #1: No

Reviewer #2: Yes

3. Have the authors made all data underlying the findings in their manuscript fully available?

Reviewer #1: Yes

Reviewer #2: Yes

4. Is the manuscript presented in an intelligible fashion and written in standard English?

Reviewer #1: Yes

Reviewer #2: Yes

5. Review Comments to the Author

Reviewer #1: I have read the manuscript entitle “Prevalence of congenital missing permanent teeth and explore any correlation with side and gender in a sample from Saudi subpopulation”. Although this is an interesting topic, the authors did not bring any new information to the literature. The written style and results presentation could also have been improved

One of my main concerns is that the authors observed a prevalence of 13.2%, that is too high once third molar agenesis were not take into consideration. However, this could be explained by the included patients age, which ranged from 9 to 60 years old. To have a correct diagnosis of agenesis in these age is a difficult task.

Reviewer #2: Thank you for your submission, and congratulations on your hard work.

The title is awkwardly phrased. Suggest rewording for clarity:

“Prevalence of Congenital Missing Permanent Teeth and Its Association with Side and Gender in a Saudi Subpopulation”.

Abstract: The abstract contains multiple grammatical issues and lacks clarity in the conclusion. Consider stating a more definitive takeaway. The term “most congenital missing” is not standard. Consider using “most commonly missing”.

Methodology: The methodology is sound, but the explanation of examiner calibration and statistical analysis is verbose and could be more concise. It would strengthen the paper to include inter- and intra-examiner reliability. Clarify how the classification of agenesis (mild/moderate/severe) was used in the analysis — this is mentioned but not utilized in the results or discussion meaningfully.

Results: Results are described adequately, but tables and figures should be referenced more clearly within the text. There is unnecessary repetition in reporting gender and arch differences in multiple sections. Consider using more precise language in the interpretation of statistical results (e.g., “not statistically significant” rather than “do not statistically significantly correlate”).

Discussion: The discussion is too long and would benefit from being more focused and structured (e.g., prevalence compared to regional and global studies, followed by implications, and then limitations). Some references are outdated or repeated. Update with more recent literature where available. The authors should more critically assess potential sources of bias and the retrospective design limitations.

Figures & Tables: Ensure figures are high-resolution and labeled appropriately. The figure legends are too vague. Specify which findings each figure is illustrating and provide clearer visual comparisons where applicable.

Replace terms like “congenital missing” with more standard terminology such as “congenitally missing teeth” or “tooth agenesis”. Consider using consistent terminology throughout (e.g., “hypodontia” vs. “agenesis”).

6. PLOS authors have the option to publish the peer review history of their article (what does this mean?). If published, this will include your full peer review and any attached files.

Reviewer #1: No

Reviewer #2: No

---

## [Author Response · Author response to Decision Letter 1]

25 May 2025

Point- by- Point Reply to Review Comments

Thank you for your encouraging remarks and for your suggestions, which have improved our manuscript. We were able to address all your suggestions as detailed in the point-by-point reply below:

Journal Requirements:

Comment Response

1 Please ensure that your manuscript meets PLOS ONE's style requirements, including those for file naming. Done

2 Thank you for stating the following financial disclosure:

[This research has been funded by Scientific Research Deanship at University of Ha’il - Saudi Arabia through project number <<RG-23 113>>].

This research has been funded by Scientific Research Deanship at University of Ha’il - Saudi Arabia through project number <<RG-23 113>>. The funders had no role in study design, data collection and analysis, decision to publish, or preparation of the manuscript.

3 When completing the data availability statement of the submission form, you indicated that you will make your data available on acceptance. We strongly recommend all authors decide on a data sharing plan before acceptance, as the process can be lengthy and hold up publication timelines. Please note that, though access restrictions are acceptable now, your entire data will need to be made freely accessible if your manuscript is accepted for publication. This policy applies to all data except where public deposition would breach compliance with the protocol approved by your research ethics board. If you are unable to adhere to our open data policy, please kindly revise your statement to explain your reasoning and we will seek the editor's input on an exemption. Please be assured that, once you have provided your new statement, the assessment of your exemption will not hold up the peer review process. We agree to make the data available on acceptance.

4 Please include a copy of Tables 1, 2, and 3 which you refer to in your text on pages 7, 8, 9. We include the copy of Tables 1 and 2. It appears that Table 3 was mistakenly inserted in place of Table 2. I apologize for the oversight. We correct the error and ensure the appropriate table 2 is included in the revised version.

Reviewer: 1

5 The written style and results presentation could also have been improved

We appreciate your suggestion regarding the written style and results presentation. We carefully reviewed the manuscript to enhance the clarity and coherence of the writing. Additionally, we revised the presentation of the results to ensure they are more clearly structured and easier to interpret, possibly by refining tables, figures, and descriptive summaries.

6 One of my main concerns is that the authors observed a prevalence of 13.2%, that is too high once third molar agenesis were not take into consideration. However, this could be explained by the included patients age, which ranged from 9 to 60 years old. To have a correct diagnosis of agenesis in this age is a difficult task. We acknowledge that the reported prevalence of 13.2% may appear elevated, especially considering that third molar agenesis was not included. As you correctly pointed out, the wide age range of the included patients (9 to 60 years old) may have influenced the results, particularly in the younger subset where dental development is still ongoing. We recognize that diagnosing agenesis accurately in younger patients can be challenging due to variability in tooth development and eruption times. This limitation be more clearly addressed and discussed in the revised version of the manuscript to provide better context for the reported prevalence.

Reviewer: 2

7 The title is awkwardly phrased. Suggest rewording for clarity:

“Prevalence of Congenital Missing Permanent Teeth and Its Association with Side and Gender in a Saudi Subpopulation”. Corrected

8 Abstract: The abstract contains multiple grammatical issues and lacks clarity in the conclusion. Consider stating a more definitive takeaway. The term “most congenital missing” is not standard. Consider using “most commonly missing”.

We have revised the abstract to correct the grammatical issues and improved the clarity of the conclusion by stating a more definitive takeaway. Additionally, the phrase “most congenital missing” has been replaced with the more appropriate term “most commonly missing.”

9 Methodology: The methodology is sound, but the explanation of examiner calibration and statistical analysis is verbose and could be more concise. It would strengthen the paper to include inter- and intra-examiner reliability.

We have revised the section on examiner calibration and statistical analysis to make it more concise and focused. Additionally, we have included the inter- and intra-examiner reliability results.

10 Clarify how the classification of agenesis (mild/moderate/severe) was used in the analysis — this is mentioned but not utilized in the results or discussion meaningfully.

Tooth agenesis was categorized based on literature into three groups: mild (one or two missing teeth), moderate (three to five missing teeth), and severe (six or more missing teeth). Additional explanations regarding these classifications were provided in the results section.

11 Results: Results are described adequately, but tables and figures should be referenced more clearly within the text. There is unnecessary repetition in reporting gender and arch differences in multiple sections. Consider using more precise language in the interpretation of statistical results (e.g., “not statistically significant” rather than “do not statistically significantly correlate”).

Done

12 Discussion: The discussion is too long and would benefit from being more focused and structured (e.g., prevalence compared to regional and global studies, followed by implications, and then limitations).

Done. We acknowledge that we streamlined the discussion with enhancing clarity and coherence.

13 Some references are outdated or repeated. Update with more recent literature where available.

We checked the references and added more update one. The repeated references were removed.

14 The authors should more critically assess potential sources of bias and the retrospective design limitations.

Corrected

15 Figures & Tables: Ensure figures are high-resolution and labeled appropriately. The figure legends are too vague. Specify which findings each figure is illustrating and provide clearer visual comparisons where applicable.

Done

16 Replace terms like “congenital missing” with more standard terminology such as “congenitally missing teeth” or “tooth agenesis”. Consider using consistent terminology throughout (e.g., “hypodontia” vs. “agenesis”).

We have revised the manuscript to replace non-standard terms like “congenital missing” with “congenitally missing teeth”.” We have also ensured consistent use of terminology throughout the text, by using “agenesis”.

---

## [Decision Letter · Decision Letter 1]

8 Sep 2025

Prevalence of Congenital Missing Permanent Teeth and Its Association with Side and Gender in a Saudi Subpopulation

PONE-D-25-02469R1

Dear Dr. Alshammari,

We’re pleased to inform you that your manuscript has been judged scientifically suitable for publication and will be formally accepted for publication once it meets all outstanding technical requirements.

Kind regards,

Mohmed Isaqali Karobari, BDS, MScD.Endo, Ph.D. Endo, FDS, FPFA, FICD, MFDS

Academic Editor

PLOS ONE

Additional Editor Comments (optional):

Dear Authors,

The authors have addressed all the comments and suggestions reviewers gave, and the manuscript has dramatically improved. The manuscript can be accepted for publication in its current form. I would like to congratulate the authors and wish them all the very best in their future endeavours.

Best regards and keep well

Reviewers' comments:

Reviewer's Responses to Questions

**Comments to the Author**

1. If the authors have adequately addressed your comments raised in a previous round of review and you feel that this manuscript is now acceptable for publication, you may indicate that here to bypass the “Comments to the Author” section, enter your conflict of interest statement in the “Confidential to Editor” section, and submit your "Accept" recommendation.

Reviewer #3: All comments have been addressed

Reviewer #4: All comments have been addressed

2. Is the manuscript technically sound, and do the data support the conclusions?

Reviewer #3: Yes

Reviewer #4: Yes

3. Has the statistical analysis been performed appropriately and rigorously? 

Reviewer #3: Yes

Reviewer #4: Yes

4. Have the authors made all data underlying the findings in their manuscript fully available?

Reviewer #3: Yes

Reviewer #4: Yes

5. Is the manuscript presented in an intelligible fashion and written in standard English?

Reviewer #3: Yes

Reviewer #4: Yes

6. Review Comments to the Author

Reviewer #3: The authors conducted a all the corrections according to the reviewers comments. The paper was enhanced by these corrections and provides a good knowledge regarding this topic in our dental field.

Reviewer #4: In this research, the authors studied the frequency of congenital tooth

missing and evaluated the relationship between gender and side characteristics.

The authors have addressed all comments. The manuscript can be accepted.

7. PLOS authors have the option to publish the peer review history of their article (what does this mean?). If published, this will include your full peer review and any attached files.

Reviewer #3: No

Reviewer #4: No

---

## [Editor Report · Acceptance letter]

PONE-D-25-02469R1

PLOS ONE

Dear Dr. Alshammari,

I'm pleased to inform you that your manuscript has been deemed suitable for publication in PLOS ONE. Congratulations! Your manuscript is now being handed over to our production team.

Kind regards,

on behalf of

Prof Dr. Mohmed Isaqali Karobari

Academic Editor

PLOS ONE